# Extracellular Vesicles in the Tumour Microenvironment: Eclectic Supervisors

**DOI:** 10.3390/ijms21186768

**Published:** 2020-09-15

**Authors:** Claudia Cavallari, Giovanni Camussi, Maria Felice Brizzi

**Affiliations:** 12i3T Scarl, University of Turin, 10126 Turin, Italy; cavallari.1184@gmail.com; 2Department of Medical Sciences, University of Turin, 10126 Turin, Italy; giovanni.camussi@unito.it

**Keywords:** extracellular vesicles, tumour microenvironment, EV engineering

## Abstract

The tumour microenvironment (TME) plays a crucial role in the regulation of cell survival and growth by providing inhibitory or stimulatory signals. Extracellular vesicles (EV) represent one of the most relevant cell-to-cell communication mechanism among cells within the TME. Moreover, EV contribute to the crosstalk among cancerous, immune, endothelial, and stromal cells to establish TME diversity. EV contain proteins, mRNAs and miRNAs, which can be locally delivered in the TME and/or transferred to remote sites to dictate tumour behaviour. EV in the TME impact on cancer cell proliferation, invasion, metastasis, immune-escape, pre-metastatic niche formation and the stimulation of angiogenesis. Moreover, EV can boost or inhibit tumours depending on the TME conditions and their cell of origin. Therefore, to move towards the identification of new targets and the development of a novel generation of EV-based targeting approaches to gain insight into EV mechanism of action in the TME would be of particular relevance. The aim here is to provide an overview of the current knowledge of EV released from different TME cellular components and their role in driving TME diversity. Moreover, recent proposed engineering approaches to targeting cells in the TME via EV are discussed.

## 1. Introduction

Extracellular vesicles (EV) include 100–5000 nm vesicles released by almost all cell types. Exosomes are a subclass of extracellular vesicles ranging from 40 to 150 nm derived from multivesicular bodies, and are distinct for their biogenesis from ectosomes and pre-apoptotic vesicles, as the latter originate from the budding of the cell plasma membrane [1]. Since the definition of these membrane vesicles is evolving, in the present review, they will be referred to as EV. EV are highly heterogeneous and likely reflect the physiological/pathological conditions of the cell from which they originated. EV are composed of a lipid bilayer and serve as carriers of information through the release of lipids, proteins, RNA, and DNA into target cells [2].

EV activate signalling pathways in cells they fuse or interact with by transferring specific genetic and non-genetic components [3,4]. EV are detected in the tumour microenvironment (TME), and emerging evidence suggests that they play a role in facilitating tumourigenesis by regulating different processes, including tumour growth, angiogenesis, immunity, and metastasis formation. Circulating EV have also been exploited as liquid biopsies and recognized as biomarkers for early detection, diagnosis, treatment and response to treatment in cancer patients [5,6,7].

During the last two decades, several studies have explored EV function in tumour from different origins, however, so far, the precise EV role remains uncertain. In cancer, EV from different cells of origin have been largely described as tumour promoters, however, available data also suggest that EV retain anti-tumour properties and can also act to restrain disease progression [8].

The tumour microenvironment (TME) contains cancer cells, displaying different phenotypes and genetic features and a number of different cell types, including stromal cells, mesenchymal cells, endothelial cells (EC), and cancer associated fibroblasts (CAF) [9,10]. Diverse immune cell subtypes can also be found in the TME, which include dendritic cells (DC), B-lymphocytes, T-lymphocytes, natural killer (NK) cells, and macrophages [9,10]. All these cells shed EV, and contribute to TME diversity [11] (Figure 1). Moreover, since EV released within the TME likely contribute to the heterogeneity of circulating EV, they have also been considered to be a fingerprint of the tumour.

Novel EV engineering approaches have offered the opportunity to control primary tumours and metastatic diseases [12,13]. Therefore, the potential application of EV as naturally delivery system for therapeutic agents has been actively explored [12,13]. The aim of this review is to give an overview of the current knowledge on EV functional diversity in the tumour setting, with particular emphasis on their role as pro- or anti- cancer intermediaries. Moreover, recent engineering approaches to move towards EV clinical application are discussed.

## 2. EV Pro-Tumourigenic Properties

The TME drives pro-tumourigenic effects by boosting tumour expansion and metastatic spread. A number of studies have suggested that EV within the TME act as central mediators of angiogenesis, immune modulation, and metastatic spread [14]. This implies that EV can be considered to be potential targets for a new generation of pharmaceuticals directed at reprogramming the TME [15,16]. Herein, the most relevant pro-tumour actions of EV released from different cell types in the TME are examined.

### 2.1. Cancer Stem Cell-EV

Tumour lesions comprise multiple subpopulations of cells, including those endowed with “stemness” properties and named cancer stem cells (CSC). CSC are responsible for tumour initiation, metastasis formation, resistance to conventional therapies and disease recurrence [17]. These relatively rare cells, also defined as cancer initiating cells (CIC), are characterized by self-renew capability, multipotency, and tumourigenicity. It has been extensively reported that CSC can communicate within the TME via EV, thus contributing to maintaining tumour heterogeneity [18].

Recent studies have shed light on the role of CSC-derived EV as tumour promoters by regulating the TME remodelling. It has been found that colon cancer stem cells expressing the extracellular matrix metalloproteinase inducer (EMMPRIN), also named CD147, are able to release CD147+ EV, which activate a signalling cascade in recipient cells boosting their migratory phenotype. Moreover, the expression of CD147 was also found to regulate EV shedding. Consistently, EV release and their downstream effects on target cells were hampered by knocking-down CD147 or by using the anti-CD147 blocking antibody [19].

EV conveying CSC signature have been recently proposed as tumour biomarkers and potential CSC targets [20]. It has been reported that EV carrying CSC-surface markers boost CSC independent crosstalk with the host and neighbouring non-CSC [21]. This translates in the local non-CSC reprogramming and the formation of distant metastatic niches [21].

It has also been reported that EV enriched in Wnt3a are released from high grade lymphoma stem cells and can reprogram the TME by activating the Wnt pathway. This translates in the expansion of the tumour bulk [22]. It has been demonstrated that diffuse large B-cell lymphomas possess a self-organized infrastructure, encompassing cells belonging to the side (SP) and non-SP population. The transition between the two populations is under the control of EV and the reciprocal expression of Wnt3a in SP cells-derived EV, and the secreted frizzled-related protein 4 in EV from non-SP cells controls this balance by regulating DNA methylation [22].

Wnt signalling plays an important role in many biological processes such as growth, development, metabolism, and stem cell maintenance [23]. The abnormal activation of the Wnt pathway is closely related to the development of tumours and controls CSC self-renewal and differentiation [24]. Recent studies have shown that EV can regulate the Wnt pathway in recipient cells. It has been found that, in colorectal cancer (CRC), EV released from fibroblasts activate the Wnt signalling pathway, allowing CRC cells to acquire stem cell properties and thus increasing the bulk of CSC [25].

Moreover, it has been found that signals driving CSC differentiation into non-stem tumour cells (non-CSC) is a bidirectional and dynamic process. EV released in the TME by CSC concur to the dynamic shift between the differentiated and undifferentiated state of cancer cells, the so-called tumour plasticity [26].

It has also been reported that CSC can evade the immune surveillance and maintain their quiescent and dormant state through their immunomodulatory properties [27]. They drive neoplastic growth and recurrence, even after long latency. Moreover, CSC, due to their ability to modulate and shape immune responses, represent relevant mediators of resistance to immunotherapeutic approaches in cancer patients [28]. At this regard, it has been reported that renal 105 + CSC-EV drive immune-escape by targeting monocyte-derived DC [29]. HLA-G+ positive EV, which were found to be crucial regulators of immune tolerance in pregnancy, also act in cancer [30]. Indeed, 105+ renal CSC-EV impair DC maturation and T cell activation by a mechanism involving the HLA-G [31].

This suggests that further efforts to dissect the mechanisms regulating the immunological profile of CSC and their released, EV as well as their crosstalk with immune cells within the TME, would be useful to rationally design immunotherapeutic interventions to eradicate malignancy.

### 2.2. Stem Cell-EV

Particular attention has been devoted to EV derived from stem cells, since they are generally involved in the network of inputs enabling cells to proliferate and migrate to sites of tissue injury [32,33]. Stem cell-derived EV can act in an autocrine manner to influence stem cell proliferation, mobilization, differentiation, angiogenesis and self-renewal [34]. Hence, it has been postulated that stem cells and EV may work synergistically in tissue repair processes, while EV dysregulation may induce microenvironment changes, leading to the loss of tissue homeostasis and the occurrence of diseases, including cancer [35].

The formation of the tumour vascular network is crucial to supply nutrients and oxygen, to remove waste products, and to provide the soil for the homing of immune cells [36]. Angiogenic factors carried by EV released from different cells, and particularly from stem cells, are receiving particular attention [37,38]. It has been reported that mesenchymal stem cells (MSC), through their released EV, promote tumour progression by boosting vascularization, and by driving changes within the TME [39,40]. Indeed, MSC-EV interact with multiple cell types in the TME to support tumour growth. Moreover, MSC-EV have the potential to elicit different cellular responses in a variety of cells through the delivery of their molecular cargo [41]. Recent evidence demonstrated that treatment with MSC-EV increases the numbers of migrated cells and the length of tubes formed by human-derived EC [42]. In addition, in vitro experiments revealed that, although MSC-EV unveil significantly lower levels of angiogenic growth factors than their conditioned media (MSC-CM), they were much more effective in promoting angiogenesis [42]. This further confirms that the entire EV cargo most likely contributes to the paracrine effects of MSC [43,44]. Moreover, it has been demonstrated that adipose mesenchymal stem cell-derived EV (ASC-EV), enriched in angiogenic factors such as Milk fat globule-EGF factor 8 (MFG-E8), Angiopoietin Like 1 (ANGPTL1), thrombopoietin and matrix metalloproteinases (MMPs) ease endothelial cell migration and activate a number of signalling pathways, resulting in the growth of the tumour vascular tree [44,45]. Besides their role in promoting angiogenesis, stem-cell-derived EV contribute to tumour progression by directly acting on tumour cell growth, migration, and invasion. It has been reported that EV released from human umbilical cord-derived mesenchymal stem cells (hUC MSC EV) promote the invasive and migratory potential of breast cancer cells by fostering ERK-mediated epithelial–mesenchymal transition (EMT) [46]. hUCMSC-EV also contribute to the in vivo growth of lung adenocarcinoma cells (LUAD). The authors demonstrated that hUCMSC-EV exert their biological effects by increasing proliferation and decreasing apoptosis. miR-410 enriched in hUCMSC-EV is crucial for their biological action, as it regulates the expression of PTEN [47].

The pro-tumourigenic action of MSC-EV has also been reported to contribute to MCF7 cell migration. The up-regulation of several cancer-related signalling pathways was found to be associated to the MCF7 cell migratory phenotype. The Wnt signalling cascade was found to be among the most relevant pathways activated in response to MSC-EV challenge [48].

It has also been reported that MSC-EV control the expansion of nasopharyngeal carcinoma (NPC) cells. It has been noticed that MSC-EV taken up by NPC cells drive proliferative and migratory cues. After MSC-EV administration, tumour cells showed significant changes in the expression of EMT markers [49]. In this regard, it has been reported that MSC-EV control the expression of the fibroblast growth factor (FGF) family member, FGF19, which on turn induces NPC cell proliferation, migration and invasion, by activating the FGFR4 signalling cascade [49].

MSC-EV also play a role in metastasis formation by contributing to tumour cell homing at distant sites. It has been reported that a more efficient tumour growth and metastasis formation occurs when myeloma cells are administered together with MSC-EV [50]. Furthermore, MSC-EV regulate self-renewal, inhibit differentiation of haematopoietic stem cells (HSC) and exert immunomodulatory action by activating neutrophils and inhibiting the DC, NK, B, and T-cell proliferation of [51]. This implies that the immune regulatory processes in the TME are strictly controlled by MSC-EV [52]. Moreover, MSC through their released EV can also exert immunosuppressive effects in solid tumours, by acting on any component of the immune system [53]. In this regard, it has been shown that, although less effective than tumour-derived EV, MSC-EV are able to modulate the recall immune responses by upregulating the expression of IL-10, TGF-β and Foxp3 [54].

Of interest, and similar to MSC, the human liver stem cells (HLSC), a mesenchymal stromal cell-like population resident in the adult liver, display immunomodulatory properties by a mechanism dependent on prostaglandin E2 (PGE2) and indoleamine 2,3-dioxygenase activity [55].

The crosstalk between MSC-EV and cancer cells in the haematopoietic niche has also been investigated. It has been reported that in vivo, the growth of multiple myeloma (MM) cells is supported by the release of MSC-EV [56]. Moreover, EV-mediated angiogenesis and osteolytic activity were found to be crucial for MM metastatic spread. In fact, MSC-EV contribute to the formation of distant pre-metastatic niches, MM cell engraftment as micro-metastases, and their further expansion as macro-metastases [56].

The list of MSC-derived “messengers” is expanding, and different molecules have been reported to act as drivers of cancer progression in a paracrine fashion [57,58]. A recent study has suggested that genetic information (mRNA and/or microRNA) can be effectively transferred from MSC to cancer cells, via EV revised in [59].

### 2.3. CAF-EV

As one of the most abundant TME components, CAF, and particularly their released EV, play key roles during tumour expansion and metastasis formation [60]. Multiple classes of molecules, including growth factors, cytokines, proteases, and extracellular matrix proteins, carried by CAF-EV mediate stroma-tumour-cell interaction [61]. Several studies sustain the role of EV derived from CAF primary cell lines on tumour proliferation, survival, migration, and invasion [62,63]. A significant role of CAF-EV in promoting the migration and invasion of oral squamous cell carcinoma (OSCC) cells has been reported [64]. It has been demonstrated that CAF-EV significantly induce migration and invasion of OSCC cells and promote dissemination of the HSC-3 (a human metastatic tongue squamous carcinoma cell line) cells in the 3D organotypic assay. Furthermore, gene profiling revealed that CAF-EV induce the expression of genes linked to tumour invasion and genes involved in several pathways associated with the tumour metabolism [64]. CAF-EV also display pro-tumourigenic effects via the activation of the PI3K/Akt and MAPK/Erk signalling in endometrial cancers [65].

CAF-EV have also been involved in the regulation of intercellular communication mechanisms leading to secondary organ localization in salivary adenoid cystic carcinoma (SACC). The authors have shown that CAF-EV are crucial for lung pre-metastatic niche formation and hence for the increased number of lung metastases [66]. The role of CAF-EV in regulating lung pre-metastatic niche formation has been linked to their ability to activate resident fibroblasts. The activation of TGF-β-mediated pathway was found to be relevant for CAF-EV-mediated matrix remodelling [67]. CAF-EV-mediated tumour aggressiveness has also been reported in pancreatic ductal adenocarcinoma (PDA). The proteomic signature of the stromal components of PDA identified the annexin A6/LDL receptor-related protein 1/thrombospondin 1 (ANXA6/LRP1/TSP1) as crucial for tumour cell crosstalk within the TME. The delivery of ANXA6/LRP1/TSP1 via CAF-EV (ANXA6 + EV) was associated with PDA aggressiveness. Moreover, since ANXA6 + EV were detected in the serum of PDA patients, ANXA6 + EV have also been proposed as PDA biomarker [68].

CAF-EV can also promote the progression of breast cancers through the transfer of their miRNA cargo. It has been demonstrated that, upon the transfer of miR-21, miR-378e, and miR-143 via CAF-EV, breast cancer cells (BT549, MDA-MB-231, and T47D cell lines) acquire an aggressive phenotype. The authors have shown that breast cancer cells treated with CAF-EV are connoted by the increased expression of stem cell and EMT markers, and acquire an anchorage-independent phenotype [69].

The tumour-promoting actions of CAF-EV, as well as the molecular mechanisms involved in these processes, have also been investigated in CRC. Depth biotype characterization of non-coding RNA (ncRNA) was analysed by next generation sequencing and bioinformatics, and revealed significant differences in CAF-EV (ncRNA) content when compared to CAF and EV derived from normal fibroblasts. In addition, since the ncRNA regulatory elements were found to be specifically packaged in CAF-EV, it has been proposed that CAF and CRC cells and/or stromal cells specific cross-talk may take place thanks to EV [70]. Studies on osteosarcomas demonstrated that the transfer of CAF-EV to cancer cells promotes their migratory and invasive capability. The enrichment of miRNA-1228 in CAF, in their secreted EV and in recipient tumour cells has been linked to the downregulation of the endogenous suppressor of cancer cell invasion (SCAI), a novel and highly conserved protein involved in invasiveness of osteosarcoma cells [71].

As supported by these data, CAF via EV contribute to many aspects of cancer development and progression by orchestrating a cell-to-cell signalling network involving different TME components [72]. Therefore, understanding how CAF-EV communicate within the TME would be particularly relevant to develop novel therapeutic approaches and/or original non-invasive diagnostic, prognostic, and predictive methods for clinical application in cancer patients.

### 2.4. EV-Mediated Tumour-EC Bidirectional Crosstalk

Likewise, CAF and EC are essential TME components [73,74]. Several proteins in EV have been implicated in the regulation of EC functions [75]. Additionally, in tumours, EC fate results from a bi-directional crosstalk with tumour cells, mainly controlled by tumour-derived EV enriched in functional proteins and miRNAs. As an example, Huang et al. [76] demonstrated that HepG2-derived vasorin (VASN, a type I transmembrane protein) induces EC migration via their released EV. Several groups reported the pro-angiogenic effects of tumour-derived EV in a variety of cancers, including glioblastoma, leukaemia, MM, melanoma, ovarian and breast cancers [77,78,79,80,81,82]. The crosstalk between tumour cells and EC has been deeply investigated in glioblastoma tumours, since aberrant vascularization is a common feature of this brain neoplasm [83]. EV from tumour cells can convey messages to normal stromal cells, in order to support angiogenesis [84]. At this regard, it has been reported that EV derived from the U251 glioblastoma cell line promote new blood vessel formation (human brain microvascular endothelial cells) by stimulating cell proliferation, motility, and tube-like structures in a dose-dependent manner. Moreover, the EV molecular characterization revealed that EV are fully equipped with pro-angiogenic mediators, such as proteolytic enzymes (gelatinases and plasminogen activators), pro-angiogenic growth factors (VEGF and TGFβ), as well as the C-X-C chemokine receptor type 4 (CXCR4) [83].

Several miRNAs enriched in EV are thought to be specifically involved in tumour angiogenesis reviewed in [85]. For example, in colorectal cancer, miRNA-9 in tumour-derived EV induce angiogenesis by inhibiting the expression of the suppression of cytokine signalling 5 (SOCS5) [86]. Similarly, tumour-derived endothelial cells (TEC) drive angiogenesis via the release of EV [87]. miRNA-24–3p and miRNA-214 act as key regulators of in vivo TEC-EV-mediated vessel formation by regulating the Wnt/β-catenin pathway [87].

Of note, it has also been shown that, in response to chemotherapy, EC promote tumour development through the secretion of circulating miRNA-503. Using an exogenous miRNA, the authors demonstrated that EC via EV drive breast cancer cells toward a proliferative and invasive phenotype by transferring their genetic content [88]. Moreover, since a high level of circulating miRNA-503 was found in breast cancer patients receiving neoadjuvant chemotherapy, it has been proposed that, in response to unfavourable conditions (chemotherapy or radiation), EC can release circulating miRNA-503 which, in turn, reprogram the TME [88].

Additional studies revealed that HeLa cell-derived EV promote metastasis by triggering the endoplasmic reticulum (ER) stress and by breaking down the integrity of the endothelial layer. Indeed, the expression of genes involved in the ER stress was found significantly increased in EC after treatment with HeLa cell-derived EV. Knock down of the protein kinase RNA-like endoplasmic reticulum kinase prevented the down-regulation of Zonula occludens-1(ZO-1) and Claudin-5 and inhibited the ER stress in cells treated with HeLa cell-derived EV [89].

As supported by these data, tumour angiogenesis can be modulated not only by soluble angiogenic factors, but also by EV derived from different cell types in the TME. Therefore, to face cancer angiogenesis and interfere with cell-to-cell communication in the TME, EV targeting should be considered a novel frontline.

## 3. EV Anti-Tumour Properties

The anti-tumour activity of stem cells-derived EV, as well as the immunomodulatory properties of tumour-derived EV (TDE) and EV derived from immune cells, have gained attention as novel tools for EV-based anti-cancer therapies. In fact, the most promising results on EV application as anti-tumour delivery system rely on their immunomodulatory action [90,91]. Herein, the most relevant data on EV-mediated anti-tumour and immunomodulatory actions are discussed.

### 3.1. Stem Cell (SC)-EV

In cancer, stem cell-EV (SC-EV) are the most attractive therapeutic options, due to their natural tumour homing capability, and to the presence of chemokine and cytokine receptors on their surface, allowing their interaction with tumour cells [92]. Several in vivo studies have shown that SC-EV preferentially migrate and incorporate into tumours after intravenous, intraperitoneal, and intracerebral delivery [93]. Studies on SC-EV oncolytic properties are currently on-going [94]. Our group has shown that HLSC-EV are able to induce an anti-tumour response by delivering selected miRNAs. The inhibition of cell survival and tumour growth was reported as the main HLSC-EV mechanism of action [95]. In fact, HLSC-EV intra-tumour administration induced regression of ectopic tumours in SCID mice [95]. In addition, it has been shown that HLSC-EV treatment inhibits TEC-derived vessel formation by down-regulating crucial pro-angiogenic genes [96].

Suppression of tumour growth by SC-EV can be accomplished by prompting apoptosis, cell cycle arrest, inhibition of angiogenesis, or by interfering with the mechanisms involved in the metastatic spread [39]. Recently, Karaoz et al. have performed comparative in vitro experiments and demonstrated that MSC-EV do not promote cancer cell proliferation [97]. Functional in vivo studies are required to validate these data. In addition, it has been recently demonstrated that miR-143 enriched in MSC-EV can be transferred to prostate cancer cells. This translates in the post-transcriptional regulation of molecules involved in cell proliferation, migration, invasion, and tumour growth (e.g., MMP-2, MMP-9 and PC3, an anti-proliferative gene), as well as in the regulation of the trefoil factor-3 (TFF3), a promising prostate cancer biomarker [98]. Moreover, it has been reported that human ovarian cancer cell growth can be inhibited by EV derived from human adipose MSC (hAMSC). This effect relies on cell cycle arrest and the activation of apoptotic signalling in mitochondria. hAMSC-EV upregulate different pro-apoptotic signalling molecules, such as BAX, CASP9, and CASP3, and downregulate the anti-apoptotic protein BCL2. Additionally, EV RNA sequencing revealed an enrichment of miRNAs, targeting different molecules linked to cancer progression (e.g., cyclin-dependent kinase (CDK) family: CDK2, CDK4, and CDK6) and cancer-survival signalling (PIK3R, RAS, MAPK, and STAT). These molecules have been referred to as the main players of hAMSC-EV-mediated anti-cancer action [99].

### 3.2. EV Immunomodulatory Properties

In recent years, the impact of EV in regulating the adaptive and innate immunity has been extensively investigated [100]. The modulation of antigen presentation, immune activation, immune suppression, and immune surveillance can be included among EV-mediated immunological activities [101]. Besides immune cells, cancer cells secrete immunologically active EV, which are able to influence both physiological and pathological processes [101]. Tumour- and immune cell-derived EV have been shown to carry tumour antigens, which may primarily act to eradicate established tumours by CD8 + T cells and CD4 + T cells [102].

Hence, EV can be considered attractive immune tools to fight cancer. EV derived from B-cells and DC have the ability to induce antigen-specific T- and B-cell responses [103]. It has been demonstrated that DC-derived EV are equipped with all essential instruments for T cell-mediated immune responses against tumours [104]. Pre-clinical studies on DC-derived EV showed that the increased number of CD8+ T lymphocytes, IFN-γ and interleukin-2 level, and the decreased number of CD25 + Foxp3 + regulatory T (Treg) cells, interleukin-10 and TGF-β, control TME remodelling to protect the host against cancer [105].

It has been shown that, besides activating CD4+ T and CD8+ T in patients with melanoma, DC-EV can be safe and feasible for clinical application [106]. It has also been reported that granulocyte-macrophage colony-stimulating factor (GM-CSF) and ascites-derived EV, likely released by immune cells, induce strong anti-cancer T cell response in patients with advanced stage of colorectal cancers [107]. Currently several clinical trials provide promising results using DC-EV for vaccination. It has been reported that DC-EV enhance the immune response and trigger the NKG2D ligands by presenting their cargo to the antigen presenting cells (APC). This, in turn, increased the number of NK in melanoma patients [106,108]. DC-EV also induce the anti-tumour immunity in patients with advanced non-small cell lung cancer (NSCLC) by boosting NKp30-dependent NK functions [109].

Among immune cells displaying anti-tumour activity, NK cells have deserved attention. NK cells can induce the immune response impairing dissemination of solid cancers and haematological malignancies [110,111]. However, the role of their released EV is still under investigation. Both in vitro and in vivo studies demonstrated that NK cells-derived EV exert cytotoxic effects on melanoma cells by presenting both perforin and Fas ligand [112]. Recent studies reported that NK cells, isolated from the blood of healthy donors, release EV expressing distinctive NK cell markers (i.e., CD56) and containing killer proteins (i.e., Fas ligand and perforin molecules). Such EV display the ability to counteract the growth of several tumour cell lines (e.g., melanoma) and to activate the immune system [112,113]. NK cells-derived EV have also been shown to mediate the anti-tumour activity via their perforin and granzyme B content, by a cytotoxic-mediated mechanism [114]. It has also been reported that peptides carried by EV derived from mast cells can be presented to DC and stimulate specific immune responses [115]. Based on these observations, the role of NK cells-derived EV on the immune surveillance makes them promising tools for future therapeutic approaches.

Lymphocyte-derived EV are not the sole immune modulators in the TME, since TDE are also considered a valuable tool for anti-cancer immunotherapeutic approaches [116]. TDE can promote the immune response and inhibit the tumour growth [116]. TDE from lymphocytic leukaemia cells significantly decrease TGF-β1 expression in DC. In addition, DC pulsed with TDE are more effective in stimulating CD4 + T cell proliferation in vitro, Th1 cytokine secretion and tumour-specific CTL responses [104].

## 4. EV Engineering for Cancer Therapy

Engineered EV are a new frontier for anti-cancer-based therapies, particularly for tailored treatments in patients at different stages of disease [117]. Among the engineering technologies, chimeric antigen receptor (CAR)-engineered T-cell (CAR-T) and miRNA-loading approaches are the most attractive. We herein describe the recent findings and the current proposed approaches combining EV.

### 4.1. CAR-T EV Engineering

Recent studies have been focused on the chimeric antigen receptor (CAR)-engineered T-cell (CAR-T) approach for cancer immunotherapy [118,119,120]. CARs are monoclonal antibody-based recombinant receptors that provide both antigen-binding and T-cell-activating functions. Once expressed in T cells, CAR-T cells acquire potent antigen-targeted cytotoxic activity and act as “living drugs” [121]. CAR-T cell-based therapy involves the genetic modification of patient’s autologous T-cells to express specific CAR for tumour antigens, followed by their ex vivo expansion and re-infusion in patients. CARs are fusion proteins of a selected single-chain fragment variable from a specific monoclonal antibody and one or more T-cell receptor intracellular signalling domains [121]. Such T-cell genetic modification can be obtained via viral- or non-viral-based gene transfer, such as DNA-based transposons, CRISPR/Cas9 technology or electroporation-mediated mRNA transfer [122].

However, concerns related to their infusion in patients are emerging. The so-called cytokine release syndrome (CRS) is one of the most adverse event associated with CAR-T cell infusion [123]. EV have already been exploited for anti-cancer therapeutic approaches, hence, the combination of CAR-T and EV technology has been proposed [124,125]. CAR EV may have a low risk of toxicities and can be generated from healthy donors and applied as ‘off the shelf therapeutics’.

Unlike in lymphoid malignancies, in solid tumours, the CAR-T approach unveiled disappointing therapeutic effects [126]. Two core mechanisms mainly related to the intrinsic TME features account for the lack of CAR-T effectiveness in solid tumours. First, active tumour-mediated immunosuppression may limit the activity of CAR-T cells; second, functional changes in T lymphocytes after their ex vivo manipulation may reduce CAR-T cell diffusion through the ECM [127]. EV have the ability to cross biological barriers and the leaky vasculature of tumours prompts their trafficking into tumours when intravenously injected [128]. Once again, EV intrinsic properties may overcome CAR-T limitations, potentially allowing their application in solid tumours. Fu at al. demonstrated that CAR-containing EV express a high level of cytotoxic molecules and inhibit tumour growth. Moreover, compared to CAR-T cells, CAR EV do not express the programmed cell death protein 1 (PD1), and their anti-tumour effects cannot be weakened by recombinant PD-L1 treatment. In addition, in a preclinical in vivo model, CAR EV administration was found to be quite safe in preventing CRS compared to CAR-T therapy [125].

These data support the possibility that CAR EV may become an effective anti-cancer targeting approach overwhelming the limitations of current treatment modalities (Figure 2). Appropriately applying cellular and EV platforms, CAR-based treatment would be more effective and might be considered the next promising targeted-based option. Although a few CAR-T/EV combinatory studies are so far available, this challenging approach should be deeply investigated.

### 4.2. miRNA-EV Engineering

Innovative technologies have been developed in recent years, exploiting EV-miRNA-loading [129]. Since miRNAs can act as oncogenes or tumour suppressors, strategies based on their inhibition and/or replacement may represent the upcoming anti-cancer options (Figure 3). Indeed, miRNA administration was found to be effective in restoring their abnormal expression in cancers [130]. Therefore, miRNA engineered EV have been proposed as a useful approach to treat cancers and to conceivably reduce sides effects of current therapeutics [130]. Approaches that increase miRNA stability and prevent nuclease degradation are the most promising choices [131]. Currently, electroporation is the most used technique to incorporate miRNAs into EV. Although effective in destabilizing the membrane of vesicles and allowing miRNA entry, EV and miRNA aggregation is still an unsolved issue, applying electroporation [132].

A recent study investigated the use of synthetic particles to vehicle miRNAs as therapeutic drugs to suppress HepG2 growth [133]. miRNA-31 and miRNA-451a engineered plasma-derived EV showed effectiveness in promoting HepG2 apoptosis and growth suppression. It has been demonstrated that EV electroporated with miRNA-31 significantly down-regulate CDK2, commonly overexpressed in HCC. Moreover, down-regulation of the transcription factor, named specificity protein 1 (SP1), has also been reported to interfere with HepG2 apoptosis, proliferation, and invasion [133]. It has been shown that EV can also efficiently deliver miRNAs to epidermal growth factor receptor (EGFR)-expressing breast cancer cells [134]. Experimental evidence suggested that upon miRNA loading, EV derived from the human embryonic kidney cell line 293 (HEK293) are therapeutically effective in targeting cancerous tissues expressing the EGFR. Intravenously injected EV successfully deliver their loaded let-7a miRNA to EGFR-expressing breast cancer tissues in RAG2–/–mice [134].

In addition, miRNA-146b, which is lost in most glioma tumours, was found to reduce invasion and motility of glioma cells by silencing the EGFR [135]. Katakowski et al. [135] investigated the role of miRNA-146b as a potential tumour suppressor in glioma. In this study, MSC electroporated with miRNA-146b were used as EV source. A single EV intra-tumour administration five days after tumour implantation was found to be effective in reducing the tumour volume [135].

Recent evidence showed that synthetic miRNA-143 introduced into MSC cells is released by EV, and is efficiently transferred to osteosarcoma cells. The delivery of MSC-EV enriched in miRNA-143 significantly reduced migration of osteosarcoma cells [136].

Of interest, it has been reported that EV protonation can be useful to generate a pH gradient across the EV membranes to enhance miRNA loading without impairing EV cellular uptake nor inducing toxicity in vivo. The biological effect was verified by loading HEK293T-derived EV with different miRNAs and evaluating their cellular targets [137].

In order to enhance the efficacy of EV-mediated miRNA delivery, a novel system has been developed [138]. The authors successfully enhanced the entrance of designated miRNA mimics or inhibitors directly in isolated EV, by using a modified calcium chloride-mediated transfection method. The delivery of miRNAs enriched EV was found to be effective in inducing miRNA overexpression or depletion, both in vitro and in vivo [138].

Recent in vivo studies have also reported that SC-derived EV loaded with exogenous anti-tumour molecules and activated with a pro-drug successfully kill cancer cells [139]. Hence, SC-derived EV have been proposed as an attractive platform for the delivery of anti-tumour agents. This would be particularly relevant for “drugs” which should go across the blood-brain barrier (BBB) to reach the target tissue. In fact, bioengineered EV have been already described to prevent brain tumour metastasis by trafficking anti-tumour proteins across the BBB [140].

Engineered EV have also been used to directly and specifically target the mutated form of the GTPase KRAS, a key driver of pancreatic cancers [141]. EV derived from normal fibroblast-like mesenchymal cells have been engineered to carry specific KRASG12D siRNAs or shRNAs (eEV). Kamerkar et al. [141] have demonstrated that the expression of CD47 on EV enhance their retention in the circulation. The presence of CD47 on EV allows evasion from circulating monocytes and increases their half-life [141]. The authors identified the functional contribution of CD47 and Ras-induced macropinocytosis in suppressing EV clearance from circulation, and in enhancing their homing to pancreatic cancer cells, respectively. Moreover, they found that EV act as ‘single targeted agent’ delivering the RNAi to the oncogenic Kras in pancreatic tumours. This translates in the enhancement of the overall survival in all experienced models [141].

## 5. Conclusions

EV have been increasingly investigated as novel intercellular communication mediators in the TME [120,142]. Recent studies have been focused on the dual aspect of EV released within the TME, with particular attention to their ability to promote or interfere with tumour progression [143]. Table 1 summarizes the most relevant results. Among SC-derived EV, MSC-EV deserve special attention as anti-cancer tools. It has been reported that MSC inhibit tumour growth, mainly by interfering with the activation of signalling pathways associated with proliferation (AKT, PI3K, and Wnt), by downregulating the X-linked inhibitor of apoptosis protein (XIAP), and by suppressing angiogenesis [144]. However, their dualistic mechanism of action in tumours related to their pro-angiogenic properties still raises concerns on their clinical application as an anti-cancer option [145]. Likewise, since MSC biologically active materials are transferred to their released EV, the precise framing of EV functions is still challenging. This implies that to move toward MSC-EV based therapies extreme caution should be devoted to clearly define as to boost their anti-cancer properties, while removing their tumour-promoting activity [146]. Due to their ability to trigger specific immune responses, EV released from both SC and tumour cells may also be considered to be a novel tool for EV-mediated anti-cancer-based therapies [147]. Exploring EV biology, and particularly the molecular mechanisms involved in immune cell targeting, interaction, and manipulation, would likely provide significant insights into immune-recognition and therapeutic intervention in cancer [148].

Due to the relevant role of the TME in tumour development, progression, and metastasis formation, targeting different cell types supporting tumour growth by EV engineering might represent a conceivable anti-tumour strategy to track [149]. Moreover, the combined application of CAR-T cells and CAR EV platform would undoubtedly strengthen the use of CAR-based anti-cancer therapies [150].

As EV cargo includes numerous proteins, lipids, and nucleic acids, valuable information on their origin and function can be garnered by a deeper cargo analysis. Particular efforts have been devoted to characterizing EV, specifically TDE collected and isolated from liquid biopsy [151]. TDE have been proposed as biomarkers, diagnostics and therapeutics, or “theranostics” [152] (Table 2). Although the EV isolation procedure is still a matter of debate, currently, the gold standard for EV purification includes differential centrifugation [153,154]. Differentially expressed proteins and genetic materials in EV recovered from tumour patients and healthy subjects are the so far most applied approach for biomarker discovery. In particular, due to their stability, high specificity, and easy sample collection, miRNAs delivered by TDE have been proven to be a useful tool for biomarker detection [155]. Several clinical trials aimed to evaluate the potential clinical application of EV in tumours from different origin are ongoing. At present, 91 are listed (https://clinicaltrials.gov/ct2/results?cond=Cancer&term=Exosomes&cntry=&state=&city=&dist=). Biological fluids such as urine and serum are the most common proposed EV sources. Data on their impact as diagnostic or prognostic tools could be available in a few years and, should validated, their possible application in cancer patients will become part of the real world. Finally, even though only a few data on the effectiveness of radiotherapy and MSC-EV as combo approach [156] are so far available, a deep investigation of the therapeutic efficacy of EV and current treatment options should be considered as a future challenge.

However, to move towards EV, anti-cancer-based therapy several hurdles should be solved (Table 2). They include potency tests, EV scalability and a full characterization in compliance with the existing regulatory frameworks for EV-based therapeutics [157].

## Figures and Tables

**Figure 1 ijms-21-06768-f001:**
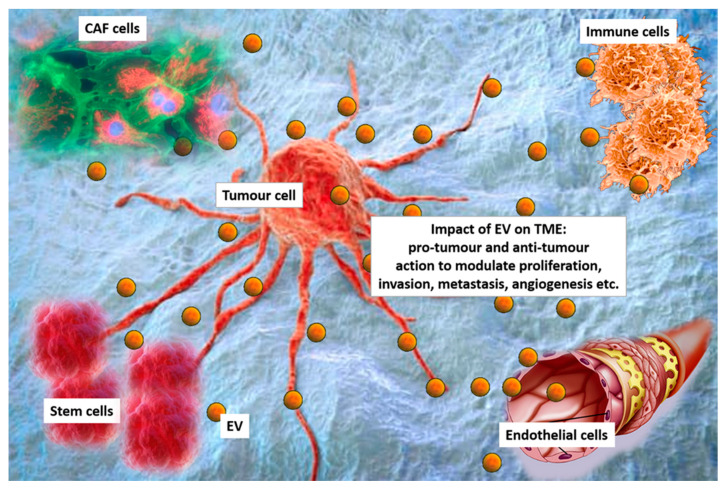
Extracellular vesicles (EV) in the tumour microenvironment (TME). Cancer cells, cancer associated fibroblasts (CAF), immune cells, stem cells and endothelial cells interact each other via EV in the TEM. EV can exert pro-tumour or anti-tumour effects to modulating proliferation, invasion, metastasis formation and angiogenesis.

**Figure 2 ijms-21-06768-f002:**
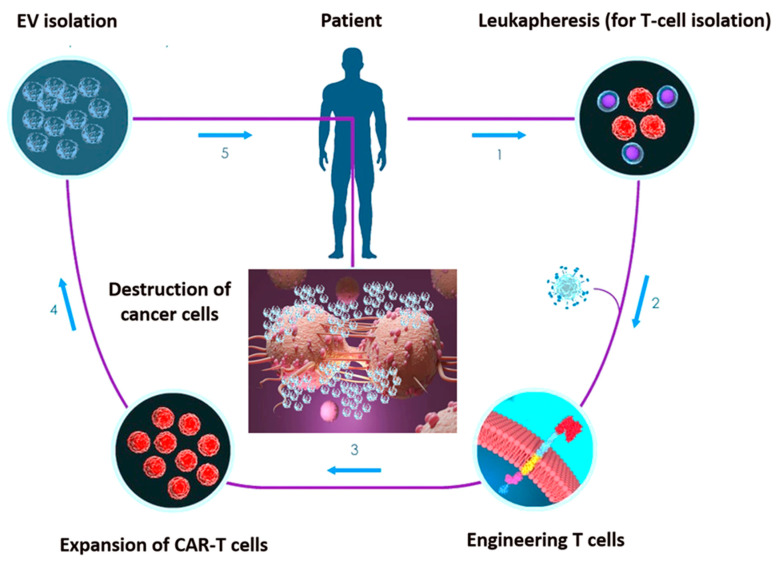
Set-up phases of chimeric antigen receptor (CAR)-engineered T-cell (CAR-T) EV engineering. 1. T cells undergo collection through positive or negative selection methods from cancer patients’ peripheral blood sample. 2. T cells are transfected with CARs through viral or non-viral transfection technology. 3. CAR-engineered T cells are ex vivo expanded in the presence of IL-2. 4. CAR-T cell-derived EV undergo isolation. 5. EV can be re-infused in the patient after chemotherapy preconditioning.

**Figure 3 ijms-21-06768-f003:**
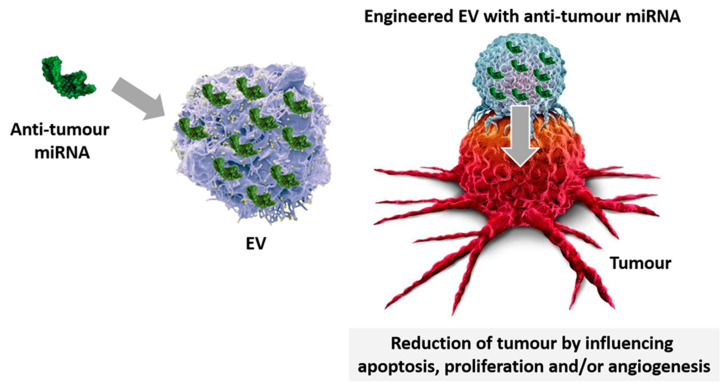
EV-miRNA-loading. Specific miRNA displaying anti-cancer properties are loaded in EV and used for treatment. Engineered miRNA-EV despatch their anti-tumour effects by regulating apoptosis, proliferation, and angiogenesis.

**Table 1 ijms-21-06768-t001:** Pro-tumour and anti-tumour functional effects of EV.

**Type of EV**	**Pro-Tumour Functional Effects**	**Ref.**
CSC-EV	boost the migratory phenotype	[19]
CSC-EV	promote formation of metastatic niches	[21]
B-cell lymphomas EV	expand tumour bulk	[22]
CRC-EV	increase CSC bulk	[25]
CSC-EV	promote transition to the undifferentiated state	[26]
renal 105 + CSC-EV	drive immuno-escape	[29]
MSC-EV	boost vascularization	[39,40]
MSC-EV	increase the number of migrated cells; increase the length of EC tube formation	[42]
ASC-EV	ease endothelial cell migration; activate the growth of the tumour vascular tree	[44,45]
hUMSC-EV	promote the invasive and migratory potential of breast cancer cells; growth of lung adenocarcinoma cells	[46]
MSC-EV	contribute to MCF-7 cell migration	[48]
MSC-EV	drive proliferative and migratory cues	[49]
MSC-EV	Immunomodulation of white blood cells	[51,52]
MSC-EV	contribute to the formation of distant pre-metastatic niches	[56]
CAF-EV	promote migration and invasion of OSCC cells; induce genes involved in tumour metabolism	[64]
CAF-EV	form pre-metastatic niche; mediate tumour aggressiveness	[66]
CAF-EV	acquire aggressive phenotype	[68]
CAF-EV	promote migratory and invasive capability of osteosarcoma	[71]
HepG2-EV	induce EC migration	[76]
Tumour-Derived EV (TDE)	support angiogenesis; form new blood vessel; stimulate proliferation and motility	[83]
TEC-EV	mediate vessel formation	[86]
TEC-EV	promote tumour development	[87]
HeLa-EV	promote metastasis	[89]
**Type of EV**	**Anti-Tumour Functional Effects**	**Ref.**
HLSC-EV	inhibit cell survival and tumour growth	[95]
MSC-EV	interfere with cancer cell proliferation	[97]
hAMSC	inhibit cancer growth; upregulate pro-apoptotic signalling molecules	[99]
DC-EV	Induce TME remodelling to protect the host against cancer	[103,104]
GM-CSF EV	induce anti-cancer T cell response in patients	[107]
DC-EV	induce anti-tumour immunity	[106,108,109]
NK-EV	exert cytotoxic effects on cancer	[112,113]
Mast cell EV	stimulate specific immune response	[115]

**Table 2 ijms-21-06768-t002:** Major benefits and drawbacks of EV treatment approaches.

Advantages and Features	Drawbacks & Issues to be Solved
Easy to use	Standardized isolation methods
Useful as biomarkers for prognostic and diagnostic purposes	Relative low loading capacity for proteins and nucleic acids
Smoothly overcome the blood brain barrier	Fine-tuning potency tests
Detectable in different biological fluids	Scalability
Confer nucleic acids (RNA) stability	The best cell source(s) to be identified
Potentially exploitable as liquid biopsy (circulating-EV)	Establish the timing and dose to use
Exploitable for specific therapeutic purposes upon engineering	Evaluate their efficacy as autologous or heterologous source

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
