# Peer review of "Extracellular Vesicles in the Tumour Microenvironment: Eclectic Supervisors"

_ijms, 2020, doi:10.3390/ijms21186768_

Round 1

Reviewer 1 Report

The present review article is a very well designed and written one 

My only comments are the followings:

(1) Are there any clinical trials and in which tumors?

(2) Is there a pre-clinical rationale regarding the combination with currently applied treatment options (e.g. chemotherapy, radiotherapy)?

(3) What re the major drawbacks in the evolution of this novel treatment approach (a Table with pros and cons might be useful for the general reader)

Author Response

Answer to Reviewer 1

We thank the Reviewer for his/her appreciation and positive comments.

The present review article is a very well designed and written one 

My only comments are the followings:

  • Are there any clinical trials and in which tumors?

Several clinical trials are ongoing. Many of them aim to evaluate the potential clinical application of EV as diagnostic and/or prognostic tools in tumor of different origin. This information and the website have been included in the present version of the MS

  • Is there a pre-clinical rationale regarding the combination with currently applied treatment options (e.g. chemotherapy, radiotherapy)?

Only a few data are so far available on these topics. According with the Reviewer suggestion we have included a sentence and the reference (ref 156) in the present version of the Ms

  • What are the major drawbacks in the evolution of this novel treatment approach (a Table with pros and cons might be useful for the general reader).

As kindly suggested by the Reviewer a Table (Table 2) has been added to the present version of the Ms.

Reviewer 2 Report

This manuscript well summarizes the work to date regarding the extracellular vescicles ruole in the tumour microenvironment. The manuscript is informative and includes a significant evaluation of published studies on the topic. To give a big picture view of the work to date, I suggest the authors to enrich the manuscript with:

  • one or more tables that provide schematic data about EV reported in the manuscript (for example EV properties, characteristics and their use)

  • argue the clinical translation of EVs as tumour therapeutic agents, for example, if there are clinical trials that use it in cancer.

Author Response

Answer to Reviewer 2

We thank the Reviewer for his/her appreciation and positive comments.

This manuscript well summarizes the work to date regarding the extracellular vescicles ruole in the tumour microenvironment. The manuscript is informative and includes a significant evaluation of published studies on the topic. To give a big picture view of the work to date, I suggest the authors to enrich the manuscript with:

  • one or more tables that provide schematic data about EV reported in the manuscript (for example EV properties, characteristics and their use)

As kindly suggested by the Reviewer a Table which summarize the most relevant effects of EV as pro-tumour or anti-tumor mediators has been included in the present version of the Ms (Table 1)

  • argue the clinical translation of EVs as tumor therapeutic agents, for example, if there are clinical trials that use it in cancer.

As kindly suggested by the Reviewer a sentence and the website to access to clinical trials has been included in the present version of the Ms.